# Polygenic risk score for acute rejection based on donor-recipient non-HLA genotype mismatch

Rui Cao[1], David P. Schladt[2], Casey Dorr[2,3], Arthur J. Matas[4], William S. Oetting[5], Pamala A. Jacobson[5], Ajay Israni[2,3], Jinbo Chen[6], Weihua Guan[1]*

1 Division of Biostatistics and Health Data Science, School of Public Health, University of Minnesota, Minneapolis, Minnesota, United States of America, 2 Hennepin Healthcare Research Institute, Minneapolis, Minnesota, United States of America, 3 Department of Medicine, University of Minnesota Medical School, Minneapolis, Minnesota, United States of America, 4 Department of Surgery, University of Minnesota Medical School, Minneapolis, Minnesota, United States of America, 5 Department of Experimental and Clinical Pharmacology, College of Pharmacy, University of Minnesota, Minneapolis, Minnesota, United States of America, 6 Department of Biostatistics, Epidemiology and Informatics, Perelman School of Medicine, University of Pennsylvania, Philadelphia, Pennsylvania, United States of America

* wguan@umn.edu

## Abstract

### Background

Acute rejection (AR) after kidney transplantation is an important allograft complication. To reduce the risk of post-transplant AR, determination of kidney transplant donor-recipient mismatching focuses on blood type and human leukocyte antigens (HLA), while it remains unclear whether non-HLA genetic mismatching is related to post-transplant complications.

### Methods

We carried out a genome-wide scan (HLA and non-HLA regions) on AR with a large kidney transplant cohort of 784 living donor-recipient pairs of European ancestry. An AR polygenic risk score (PRS) was constructed with the non-HLA single nucleotide polymorphisms (SNPs) filtered by independence ($r^2 < 0.2$) and P-value ($< 1 \times 10^{-3}$) criteria. The PRS was validated in an independent cohort of 352 living donor-recipient pairs.

### Results

By the genome-wide scan, we identified one significant SNP rs6749137 with HR = 2.49 and P-value = $2.15 \times 10^{-8}$. 1,307 non-HLA PRS SNPs passed the clumping plus thresholding and the PRS exhibited significant association with the AR in the validation cohort (HR = 1.54, 95% CI = (1.07, 2.22), p = 0.019). Further pathway analysis attributed the PRS genes into 13 categories, and the over-representation test identified 42 significant biological processes, the most significant of which is the cell morphogenesis (GO:0000902), with 4.08 fold of the percentage from *homo species* reference and FDR-adjusted P-value = $8.6 \times 10^{-4}$.

**Data Availability Statement:** The individual-level genotyping data is not public available per the informed consent. All the data of PRS SNPs

including GWAS summaries and PRS weights are listed in the supplementary information.

**Funding:** This study was supported in part by NIH/NIAID grants 5U19-AI070119 and 5U01-AI058013. The funders had no role in study design, data collection and analysis, decision to publish, or preparation of the manuscript.

**Competing interests:** The authors have declared that no competing interests exist.

**Abbreviations:** AR, acute rejection; CI, confidence interval; DeKAF, Deterioration of Kidney Allograft Function; GEN-03, Genomics of Kidney Transplantation; GO, Gene Ontology; GWAS, genome-wide association studies; HLA, human leukocyte antigens; HR, hazard ratio; IBS, identity-by-state; MAF, minor allele frequency; PRA, panel reactive antibodies; PRS, polygenic risk score; SD, standard deviation; SNP, single nucleotide polymorphism.

## Conclusions

Our results show the importance of donor-recipient mismatching in non-HLA regions. Additional work will be needed to understand the role of SNPs included in the PRS and to further improve donor-recipient genetic matching algorithms.

**Trial registry:** Deterioration of Kidney Allograft Function Genomics (NCT00270712) and Genomics of Kidney Transplantation (NCT01714440) are registered on ClinicalTrials.gov.

## 1. Introduction

Acute rejection (AR) is a major complication after kidney transplantation and is strongly associated with long-term transplant outcomes [1]. Several factors affect the risk of AR, such as recipient's age, race, donor type, and delayed graft function [2], while the role of donor-recipient genetic mismatching, is not currently well understood. Current donor-recipient matching in kidney transplant mainly focuses on blood type and SNPs within the human leukocyte antigen (HLA) region. HLA mismatching has been shown to be a significant risk factor for kidney transplant outcomes [3]. However, rejection can still occur in HLA identically matched kidney allografts [4]. Using genome-wide association studies (GWAS) [5], several recipient genetic loci have been significantly associated with transplant outcomes, such as recipient's immune response and post-transplant hyperglycemia [6, 7], but investigation if donor' genotype or donor-recipient genetic mismatching beyond the HLA region can affect transplant outcomes remains limited. Zhang et al. [8] demonstrated a significant association between renal allograft survival and the proportion of genome-shared identity-by-descent (pIBD) independent of HLA mismatches between donor-recipient pairs of similar ancestry. Pineda et al. [9] found a significantly higher number of mismatched non-histocompatibility antigen (non-HLA) variants in antibody-mediated rejection (AMR). Steers et al. [10] reported donor-recipient mismatching on *LIMS1* gene polymorphisms was associated with the rejection of kidney allografts, from a total of 50 common gene loci analyzed in the study. Not limited to the studies above, recent studies [11–13] also demonstrated that non-HLA genetic factors can also contribute to the influence on kidney post-transplant outcomes.

Our previous GWAS [14] identified several genome-wide significant SNPs where donor-recipient mismatching was associated with AR, but the SNP functions and clinical significance remain unclear. The GWAS approach for kidney transplant outcomes is often limited by sample sizes and lack of statistical power. It is likely that there exist causal SNPs outside the HLA region with moderate effect sizes, which cannot achieve the stringent genome-wide level of significance. Alternatively, the polygenic risk score (PRS) has gained its popularity by pooling many potentially associated SNPs with a relaxed threshold. Integrating genetic information from multiple genes, the PRS has been shown to predict clinical risk, such as cancer [15] or treatment response [16]. Typically, a PRS is calculated as a weighted sum of reference allele counts of a certain uncorrelated SNP set, which is usually filtered from their GWAS P-values. Following this, a validation step of the PRS is then carried out on an independent data set. Recently, the PRS has been shown to be a powerful method for predicting post-kidney transplant outcomes, such as type 2 diabetes [17] and non-melanoma skin cancer [18].

In this study, we first carried out a GWAS on AR using the Deterioration of Kidney Allograft Function (DeKAF) Genomics cohort data. Unlike a typical GWAS, the independent variable in our association study is not the reference allele count but an identity-by-state (IBS) [19] mismatch score, accounting for the donor-recipient genetic mismatch. After the weights of the SNPs were estimated in the DeKAF Genomics cohort, the PRS was then validated in another

independent cohort, the Genomics of Kidney Transplantation (GEN-03). Our study represents the first to identify a PRS with a set of autosomal non-HLA SNPs whose matching can predict the risk of AR in kidney transplant recipients.

## 2. Materials and methods

### 2.1 Data

Kidney transplant clinical information and genotyping data was collected from two cohort studies: DeKAF Genomics (2005–2011, NCT00270712) and GEN-03 (2012–2016, NCT01714440). Informed consent was obtained from all participants prior to their participation in the studies. The data were accessed for research purposes from 2021 to 2023 for this particular study. De-identified data were prepared and used in this study. Due to the limitation of informed consent, only data from GEN-03 are publicly available (dbGaP Study Accession: phs001667.v1.p1). The living donors and recipients analyzed were of European ancestry. Demographic statistics for the two cohorts are shown in Table 1.

The HLA region is defined as 25,759,242 to 33,534,827 bp on chromosome 6 [20], GRCh37. In the DeKAF Genomics cohort, genotypes of 837,930 SNPs (12,852 in HLA region) were determined with the AFR-AMR Axiom chip (Affymetrix, Santa Clara, CA) [21], while in GEN-03 cohort, genotypes of approximately 782,000 (13,326 in HLA region) variants were determined on a custom exome-plus Affymetrix TxArray SNP chip [22]. Genotype calling was performed in one batch on the Affymetrix Genotyping Console v4.0 using the GT1 algorithm, based on BRLMM-P (Affymetrix, Santa Clara, CA). Genotyping details can be found in a previous study [23]. Extensive quality control (QC) was performed on SNP genotypes according to community standards. We removed samples with a genotype missing rate >3%. Next, we selected a set of high quality autosomal SNPs (genotyping rate >99%, minor allele frequency >10%, Hardy Weinberg Equilibrium (HWE) P-value >0.001, pairwise LD ($r^2$) between SNPs <0.2, located outside regions known for long-range LD) to calculate relatedness, heterozygosity, and principal components (PCs). Samples with very high heterozygosity and suspected contamination were re-assayed and removed if high heterozygosity could not be resolved. We further removed monomorphic SNPs, and SNPs with HWE P-values <0.001 (using a subset of European samples) and/or missingness rate >5%.

**Table 1. Demographic statistics for European individuals in DeKAF Genomics and GEN-03.**

| Variable (Unit) | Mean (SD) / n (%) | |
|---|---|---|
| | **DeKAF Genomics** | **GEN-03** |
| Sample size (pairs) | 784 | 352 |
| AR events | 161 (20.5%) | 61 (17.3%) |
| Recipient age (years) | 49.3 (15.6) | 49.5 (15.2) |
| Recipient gender (male) | 521 (66.5%) | 227 (64.5%) |
| Donor age (years) | 44.1 (11.2) | 45.2 (12.0) |
| Donor gender (male) | 315 (40.2%) | 147 (41.8%)* |
| Donor-recipient gender mismatch | 344 (43.9%) | 175 (49.7%)* |
| HLA mismatch | 3.2 (1.7) | 3.3 (1.7) |
| Median follow-up time (days) | 382.5 | 456 |
| Recipient positive PRA | 447 (57.0%) | 219 (62.2%) |
| Recipient prior non-kidney transplant | 63 (8.0%) | 27 (7.7%) |

*14 donors with missing gender information.

We imputed untyped autosomal SNPs using 1000 Genomes Project phase 3 genotypes [24, 25] and Genome of the Netherlands v5 genotypes [26] as reference panels for both phasing and imputation. Genotypes were phased using SHAPEIT2 [27]; and imputed with IMPUTE2 [28].

## 2.2 Association between donor-recipient mismatching and AR

To select SNPs for the donor-recipient mismatching PRS, we first carried out a genome-wide scan using the DeKAF Genomics cohort. The outcome was defined as the first clinical AR event post-transplantation. The AR was defined by each center and was determined by each transplant center's treating physician at time of diagnosis. To account for the donor-recipient genetic mismatch on each SNP, the IBS mismatch score [19] was defined as the absolute difference of donor's and recipient's genotype on a SNP. For $i$th donor-recipient pair and $j$th variant, the IBS mismatch score is

$$M_{IBS(ij)} = |D_{ij} - R_{ij}|.$$

The association analysis was carried out through a Cox regression model with the IBS mismatch score as a covariate. Additional covariates included the recipient's age, gender, panel reactive antibodies (PRA) status, two-digit HLA matching code, and prior non-kidney transplant. Quality control (QC) procedures filtered out SNPs with a minor allele frequency (MAF) $< 0.05$, IBS mismatch score $< 0.05$ sample missingness $> 0.1$, or imputation info $< 0.8$, and only overlapped SNPs in both cohorts were analyzed. We also calculated the AR heritability as the variance proportion explained by genome-wide IBS mismatching scores and estimated it in the DeKAF Genomics cohort by Genome-wide Complex Trait Analysis (GCTA-GREML) method [29].

## 2.3 Polygenic Risk Score (PRS)

We calculated and validated the donor-recipient mismatching PRS in the GEN-03 cohort. Variant clumping was performed by PLINK 1.9 [30], with a $r^2 < 0.2$ threshold and 200 kb window size, and the SNPs with multiple P-values cutoffs ($5 \times 10^{-2}$, $1 \times 10^{-2}$, $1 \times 10^{-3}$, $1 \times 10^{-4}$, $1 \times 10^{-5}$) in the previous genome-wide scan were selected. After testing the PRS performance in the DeKAF Genomics cohort, we further proceeded the SNPs passing the $1 \times 10^{-3}$ cutoff for validation in the GEN-03 cohort. This p-value cutoff was selected given a consideration of both statistical significance in the training data and the number of SNPs. The weights of PRS were computed by the effect size estimates in the genome-wide scan, and the PRS was validated in the GEN-03 cohort under the same regression model in the genome-wide scan stage.

## 2.4 Pathway analysis

The PRS SNPs were attributed to the nearby genes by a maximum distance of 20,000 base pairs. The genes were then categorized through PANTHER 18.0 [31]. The over-representation tests on their biological functions were carried out based on PANTHER GO-Slim Biological Process database.

## 3. Results

The DeKAF Genomics cohort and GEN-03 cohorts (Table 1) contained 784 and 352 living donor-recipient pairs, respectively. All donors and recipients were of European ancestry, and all allografts were from living donors. Comparison of the cohort demographics, such as age, gender, and clinical covariates, shows that the two cohorts consist of similar populations of both donors and recipients.

In the DeKAF Genomics cohort, we estimated the AR observed-scale heritability as 0.53 with P-value = $4.18 \times 10^{-3}$. The genome-wide scan result is shown in Fig 1, and the genomic inflation factor was well controlled as 1.05. After the genome-wide scan in the DeKAF Genomics cohort, we filtered out 1,308 independent SNPs (S1 Appendix) to construct the PRS. One of the SNPs is located within the HLA region and 1,307 of the SNPs are located outside the HLA region. Since the HLA matching has already been adjusted in the genome-wide scan model, we excluded the HLA SNP from the PRS. The weights in the PRS were then computed by the estimated effect sizes (Table 2), and the PRS for each individual in the GEN-03 cohort was then computed and regressed onto the AR (PRS distribution shown in Fig 2). The estimated hazard ratio (HR) per SD for the PRS is 1.54 (95% CI: 1.07, 2.22). Because most of the donor-recipient pairs are related, we carried out a sensitivity analysis for the PRS after removing 6 unrelated donor-recipient pairs (kinship coefficient < 0.1) in GEN-03. The association is still significant for the PRS with HR = 1.70, 95% CI: (1.12, 2.35) and P-value = 0.011.

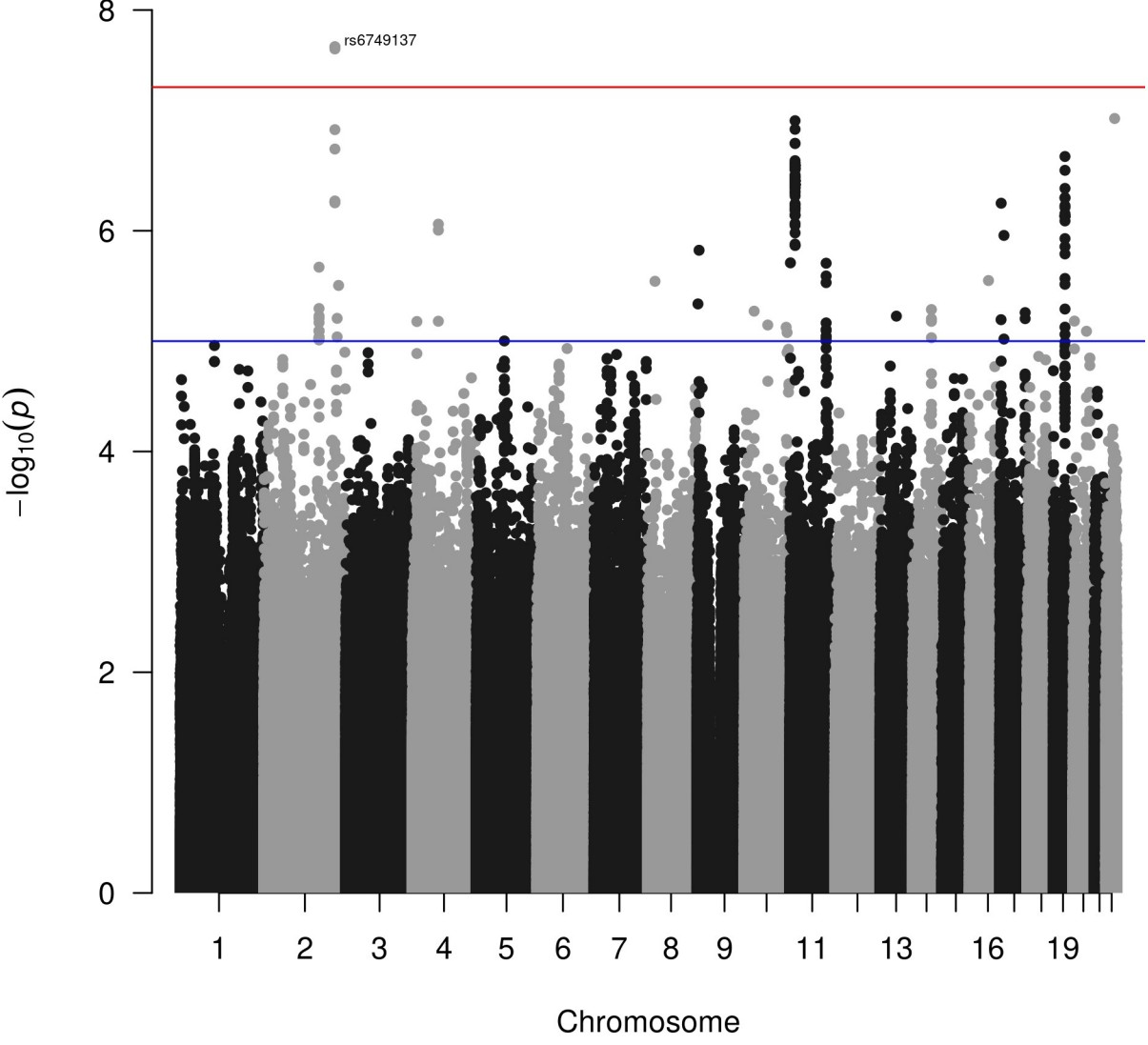

**Fig 1. Manhattan plot of the genome-wide scan in the DeKAF Genomics cohort.**

**Table 2. Top 10 weighted SNPs in the PRS.**

| SNP | Gene | Chromosome | Position (bp) | Reference allele (A0) | Alternative allele (A1) | A1 frequency | Mean IBS | Effect size | GWAS P-value |
|---|---|---|---|---|---|---|---|---|---|
| rs6749137 | *CPS1* | 2 | 211541165 | G | A | 0.073 | 0.189 | 0.913 | 2.15E-08 |
| rs9992694 | *KIAA0922* | 4 | 154535573 | G | T | 0.059 | 0.075 | 0.890 | 2.98E-05 |
| rs6557986 | *PIWIL2* | 8 | 22167613 | A | G | 0.084 | 0.051 | 0.936 | 3.06E-04 |
| rs60263168 | *PIWIL2* | 8 | 22197019 | C | CA | 0.083 | 0.052 | 1.191 | 2.87E-06 |
| rs1438457 | /* | 8 | 23558707 | G | A | 0.051 | 0.065 | 0.912 | 1.31E-04 |
| rs7898061 | /* | 10 | 133644620 | C | T | 0.058 | 0.056 | 1.059 | 2.43E-05 |
| rs12798364 | *ANO1* | 11 | 69918715 | G | A | 0.09 | 0.204 | -0.887 | 3.86E-04 |
| rs4930758 | *NTF3 ANO2* | 12 | 5636577 | G | A | 0.924 | 0.070 | 0.964 | 1.10E-04 |
| rs7208983 | *CAMKK1 P2RX1* | 17 | 3781931 | T | C | 0.081 | 0.071 | 0.922 | 8.30E-05 |
| rs190087598 | *PTPRM* | 18 | 8289510 | A | G | 0.058 | 0.148 | -1.068 | 6.15E-04 |

*No known gene attributed to the SNP.

Additionally, we conducted a pathway analysis and checked the biological functions of the SNPs in the PRS. 1,005 of the PRS SNPs were successfully mapped to their nearby genes, which can be attributed to 13 categories (Fig 3, S2 Appendix). The over-representation test on the biological process identified 42 significant GO biological processes (Fig 4), where the most significant term was the cell morphogenesis (GO:0000902) with 4.08 fold of expectation and FDR-adjusted P-value $8.61 \times 10^{-4}$.

## 4. Discussion

Donor-recipient genetic mismatching is a key risk factor of kidney post-transplant outcomes; however, our knowledge on the effect of mismatching beyond HLA region on AR risk is limited. Leveraging a genome-wide association scan in donors and recipients, our study identified a subset of independent SNPs associated with the post-transplant AR and constructed a non-HLA PRS by the weighted sums of these SNPs. Through an independent validation cohort, the

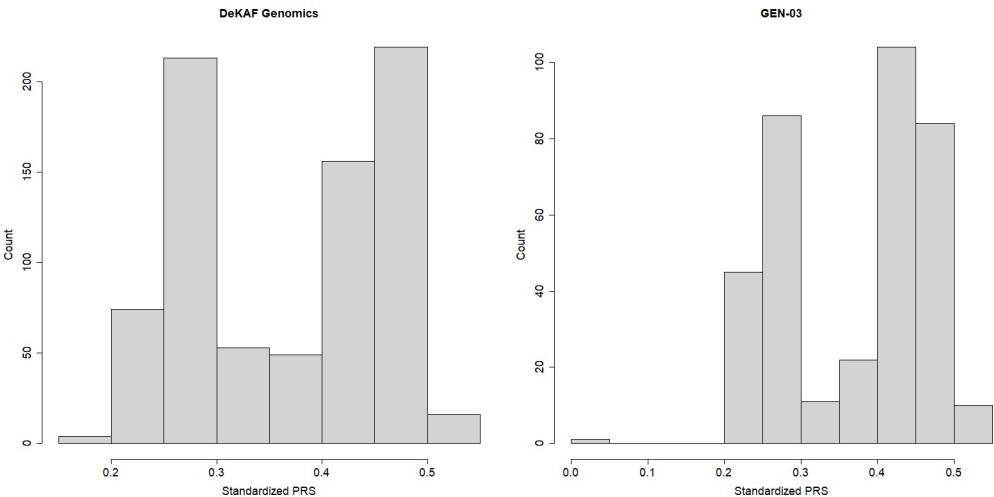

**Fig 2. The distribution of donor-recipient mismatching PRS scores in the DeKAF Genomics and GEN-03 cohorts.**

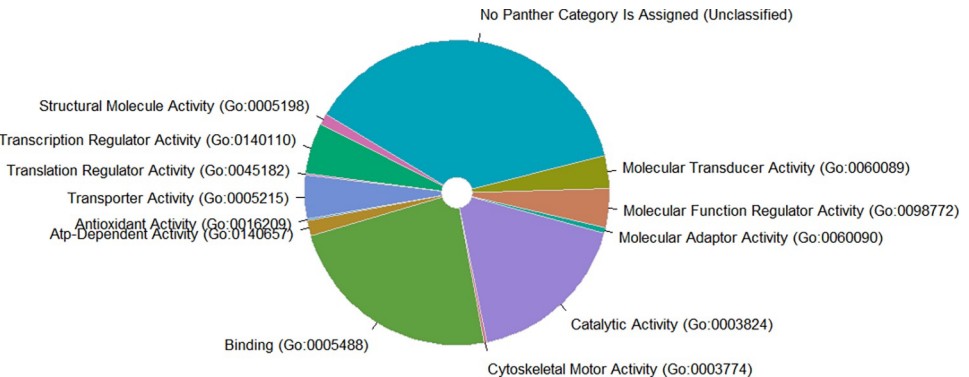

**Fig 3. Functional classification of the genes involved in donor-recipient mismatching PRS.**

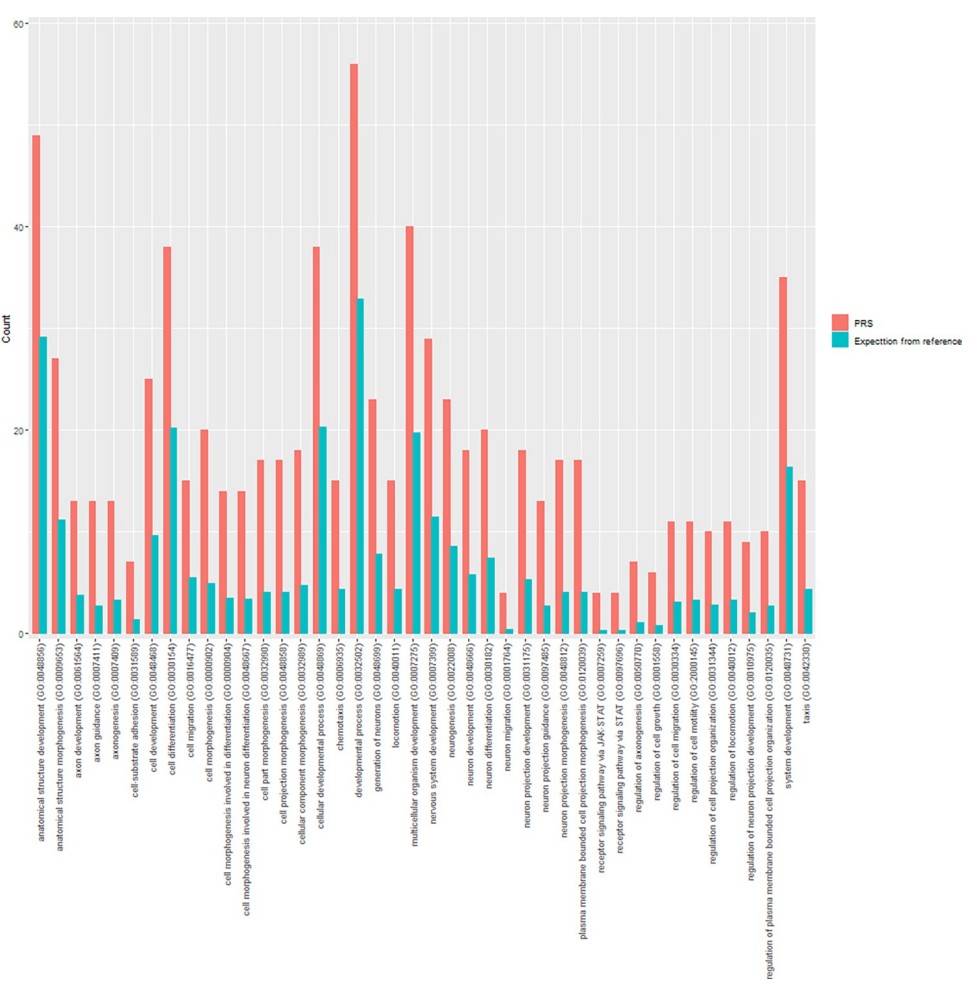

**Fig 4. Results of over-representation test for the PRS genes.**

PRS remained significantly associated with the AR after adjusting the demographics and known clinical covariates including HLA matching status.

We then identified the biological functions of the PRS SNPs. One of the top weighted SNPs, rs9992694, is an intro variant of gene *KIAA0922*, a previously reported regulator of thymocyte proliferation [32], where the T cells play a key role in T-cell mediated rejection [33]. Another top weighted SNP, rs190087598, is an intro variant of gene *PTPRM*, where another gene in the protein tyrosine phosphatase (PTP) family, *PTPN22*, has been reported significant in kidney AR [34]. The pathway analysis identified 42 significant GO biological processes. The most significant term, cell morphogenesis (GO:0000902), was previously reported specifically significant in the cells from the normal allograft [35], showing its potential impacts on AR development on the basis of normal kidney transplantation.

Despite our findings on the two SNPs above, our knowledge on the functions of rest of the SNPs in the PRS remains limited, confounded by the fact that approximately 23% of the SNPs cannot be mapped to any known genes. Although the PRS is believed to integrate moderate signals that cannot be detected by the stringent GWAS Bonferroni thresholds, currently we have limited understanding on how these SNPs likely contribute to the AR prediction. Also, it should be noted that due to sample size limitation of other ancestries, our study only analyzed patients of the European ancestry. Further multi-ancestry studies will be required for validating our results or developing population specific PRS.

Using data from two independent cohorts, our study identified a set of non-HLA PRS SNPs and validated the PRS prediction on the risk of AR. Our results showed evidence of donor-recipient matching mechanism beyond our current knowledge on the HLA region. After further validation, the PRS can be a future clinical predictor of the AR risk and provide an additional information for preoperative donor-recipient matching and selection of immunosuppression.

## Supporting information

**S1 Appendix. 1,308 independent SNPs selected by genome-wide scan.**
(XLSX)

**S2 Appendix. Over-representation test on PRS genes using reference of PANTHER GO-Slim biological processes.**
(XLSX)

## Acknowledgments

The authors acknowledge the Minnesota Supercomputing Institute (MSI) at the University of Minnesota for providing resources that contributed to the research results reported within this paper. URL: http://www.msi.umn.edu.

## Author Contributions

**Conceptualization:** Rui Cao, David P. Schladt, Casey Dorr, Arthur J. Matas, William S. Oetting, Pamala A. Jacobson, Ajay Israni, Jinbo Chen, Weihua Guan.

**Data curation:** Rui Cao, David P. Schladt, Casey Dorr, Arthur J. Matas, William S. Oetting, Pamala A. Jacobson, Ajay Israni, Weihua Guan.

**Formal analysis:** Rui Cao.

**Funding acquisition:** David P. Schladt, Casey Dorr, Arthur J. Matas, William S. Oetting, Pamala A. Jacobson, Ajay Israni, Weihua Guan.

**Investigation:** Rui Cao, Weihua Guan.

**Methodology:** Rui Cao, Weihua Guan.

**Project administration:** Rui Cao, David P. Schladt, Casey Dorr, Arthur J. Matas, William S. Oetting, Pamala A. Jacobson, Ajay Israni, Weihua Guan.

**Resources:** Rui Cao.

**Software:** Rui Cao.

**Supervision:** Weihua Guan.

**Validation:** Rui Cao.

**Visualization:** Rui Cao.

**Writing – original draft:** Rui Cao, Weihua Guan.

**Writing – review & editing:** Rui Cao, David P. Schladt, Casey Dorr, Arthur J. Matas, William S. Oetting, Pamala A. Jacobson, Ajay Israni, Jinbo Chen, Weihua Guan.

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
