## [Decision Letter · Decision Letter 0]

4 Dec 2023

PONE-D-23-29389Polygenic Risk Score for Acute Rejection Based on Donor-Recipient non-HLA Genotype MismatchPLOS ONE

Dear Dr. Guan,

Thank you for submitting your manuscript to PLOS ONE. After careful consideration, we feel that it has merit but does not fully meet PLOS ONE’s publication criteria as it currently stands. Therefore, we invite you to submit a revised version of the manuscript that addresses the points raised during the review process.

We consider that the manuscript lacks precision in various aspects, and the data presented need to be completed and strengthened. We invite you to submit a revised version of the manuscript that addresses all the points raised during the review process (see below Reviewer's comments). 

We look forward to receiving your revised manuscript.

Kind regards,

Prof. Pierre Bobé

Academic Editor

PLOS ONE

“This study was supported in part by NIH/NIAID grants 5U19-AI070119 and 5U01-AI058013. The grants were awarded to David P. Schladt, Casey Dorr, Arthur J. Matas, William S. Oetting, Pamala A. Jacobson, Ajay Israni, Weihua Guan”

“This work was supported by the National Institutes of Health. Grant numbers 5U19-AI070119 and 5U01-AI-58013. The authors also acknowledge the Minnesota Supercomputing Institute (MSI) at the University of Minnesota for providing resources that contributed to the research results reported within this paper. URL: http://www.msi.umn.edu”

“This study was supported in part by NIH/NIAID grants 5U19-AI070119 and 5U01-AI058013. The grants were awarded to David P. Schladt, Casey Dorr, Arthur J. Matas, William S. Oetting, Pamala A. Jacobson, Ajay Israni, Weihua Guan”

Reviewers' comments:

Reviewer's Responses to Questions

**Comments to the Author**

1. Is the manuscript technically sound, and do the data support the conclusions?

Reviewer #1: Partly

Reviewer #2: Yes

2. Has the statistical analysis been performed appropriately and rigorously? 

Reviewer #1: No

Reviewer #2: Yes

3. Have the authors made all data underlying the findings in their manuscript fully available?

Reviewer #1: No

Reviewer #2: No

4. Is the manuscript presented in an intelligible fashion and written in standard English?

Reviewer #1: Yes

Reviewer #2: Yes

5. Review Comments to the Author

Reviewer #1: In this manuscript, Cao et al reported associations between donor-recipient polygenic risk and acute rejection in two donor-recipient pairs of European ancestry. They first carried out a GWAS for time to AR, selected SNPs of interest to generate a PRS (within and/or outside the HLA region) and applied it to an external donor-recipient cohort. They reported significant associations between the non-HLA PRS with acute rejection in both cohorts.

I have several concerns about this study.

1. In the introduction, the authors state that several genome-wide studies have reported associations between D-R genetic mismatches and graft outcomes. In particular, the study from Zhang et al reported associations between D-R pIBD and graft outcomes. The authors of this manuscript propose to study an IBS PRS, without adding much novelty to the current literature.

2. The IBS PRS was constructed using arbitrary p value and r2 threshold. It might be useful to create different PRSs using several pruning parameters. The top 10 SNPs included in the PRS that are shown in table 2 have a MAF of less than 10%, and only 4 SNPs reached the genome-wide significance. Given the relatively low number of DR pairs, there is a concern that this study is not statistically powered, and the associations might be spurious. In addition, since this is a genome-wide association study, the authors should add lambda calculation, heritability estimates, as well as the Manhattan plot of the genome-wide significant signals for a quality check. The Authors might consider imputing genotype data to allow for higher SNP coverage and power.

3. Despite several studies showing that HLA mismatch is a strong predictor of AR, the Authors found no significant association. This may represent a concern regarding the validity of the PRS. The Authors should expand a discussion on this finding.

Reviewer #2: The manuscript describes an interesting approach to transplantation matching, PRS based on non-HLA genetics of recipients and donors of kidney tx. It is well written and results are sound. I have two major points.

The first one is, I'd assume, easy to correct: only some of the papers published on genomics of kidney transplantation have been cited. It would be fair (and correct scientific conduct) to make a more comprehensive list, missing appears at least Reindl-Schwaighofer et al 2019, Markkinen et al 2022, and Sun et al 2023. I do agree they are not on PRS but are important steps in the field.

The second one is something that I see too often: there is no access to or descriptions of data analysis details at all. For example, the github portal, or similar, would be a good way to show the code and how analyses were conducted. We usually do not accept laboratory reports without sufficient descriptions of methods (not always very detsiled, agreed!), in the same way reports heavily based on data analysis should show how the data analyses were done.

6. PLOS authors have the option to publish the peer review history of their article (what does this mean?). If published, this will include your full peer review and any attached files.

Reviewer #1: No

Reviewer #2: No

---

## [Author Response · Author response to Decision Letter 0]

2 Feb 2024

Reviewer #1: In this manuscript, Cao et al reported associations between donor-recipient polygenic risk and acute rejection in two donor-recipient pairs of European ancestry. They first carried out a GWAS for time to AR, selected SNPs of interest to generate a PRS (within and/or outside the HLA region) and applied it to an external donor-recipient cohort. They reported significant associations between the non-HLA PRS with acute rejection in both cohorts.

I have several concerns about this study.

1. In the introduction, the authors state that several genome-wide studies have reported associations between D-R genetic mismatches and graft outcomes. In particular, the study from Zhang et al reported associations between D-R pIBD and graft outcomes. The authors of this manuscript propose to study an IBS PRS, without adding much novelty to the current literature.

Authors’ reply: The authors appreciate the feedback and hope to clarify. Unlike previous studies, our study is the first one identifying a set of SNPs from the whole genome whose matching can predict AR risk. Specifically, Zhang et al. leveraged an unsupervised approach to study AR in a single cohort, with their main focus on the proportion of IBD which represents donor-recipient relatedness. In contrast, our study involved two independent cohorts and leveraged a supervised approach based on clumping and thresholding PRS. We successfully identified and validated a subset of SNPs from the whole genome whose IBS matching are potentially associated with the AR risk to create the polygenic risk score. The PRS SNPs are worth further study to investigate their roles in AR occurrence. To make the novelty of our study clearer, we added a sentence at the last of Introduction section, addressing the novelty of our study compared to previous ones.

2. The IBS PRS was constructed using arbitrary p value and r2 threshold. It might be useful to create different PRSs using several pruning parameters. The top 10 SNPs included in the PRS that are shown in table 2 have a MAF of less than 10%, and only 4 SNPs reached the genome-wide significance. Given the relatively low number of DR pairs, there is a concern that this study is not statistically powered, and the associations might be spurious. In addition, since this is a genome-wide association study, the authors should add lambda calculation, heritability estimates, as well as the Manhattan plot of the genome-wide significant signals for a quality check. The Authors might consider imputing genotype data to allow for higher SNP coverage and power.

Authors’ reply: The authors sincerely appreciate the valuable feedback. We agree with the concern of the power of our study given the high dimensionality of GWAS data. We take the PRS approach in hope to include SNPs that have moderate effect sizes and cannot be detected using the stringent GWAS p-value cutoff. For the GWAS part, we updated our analysis by adding an additional SNP QC: mean IBS mismatch score > 0.05 to avoid extreme values in the GWAS results, similar to the GWAS filtering of low MAF SNPs. Additionally, the number of 2-digit HLA mismatches was also adjusted in the GWAS model. The GWAS results including lambda calculation, heritability estimates, and a Manhattan plot were added to the main text. We observed little inflation of the test statistics with a genomic control lambda of 1.05.

We agree that the imputed genotype may improve the statistical power. In fact, we used the imputed genotype data and applied an imputation info > 0.8 filter to QC the SNPs. To avoid confusion, we now added more detailed imputation information in the Method section. 

3. Despite several studies showing that HLA mismatch is a strong predictor of AR, the Authors found no significant association. This may represent a concern regarding the validity of the PRS. The Authors should expand a discussion on this finding.

Authors’ reply: Thank you for the feedback provided. We acknowledge the HLA mismatching is highly related to AR. We hypothesize that the nonsignificant association between HLA mismatch and AR is likely due to: 1). the donors and recipients in our cohorts are already HLA-matched since that is part of donor selection in the clinic, and 2). we adjusted the 2-digit HLA matching in our GWAS model. In our updated GWAS result, there is only one HLA SNP rs9380207 that can potentially be included in the PRS. The P-value of the HLA SNP was 1.73×10^(-4) and did not pass the genome-wide significance. We excluded this SNP to create a non-HLA PRS specifically. We have clarified our method and the reason for excluding the non-HLA SNP in the Results section.

Reviewer #2: The manuscript describes an interesting approach to transplantation matching, PRS based on non-HLA genetics of recipients and donors of kidney tx. It is well written and results are sound. I have two major points.

1. The first one is, I'd assume, easy to correct: only some of the papers published on genomics of kidney transplantation have been cited. It would be fair (and correct scientific conduct) to make a more comprehensive list, missing appears at least Reindl-Schwaighofer et al 2019, Markkinen et al 2022, and Sun et al 2023. I do agree they are not on PRS but are important steps in the field.

Authors’ reply: The authors appreciate the comment and the mentioned reference. We acknowledge the significance of their discoveries on non-HLA matching and renal post-transplant outcomes and added them into citation list of this paper.

2. The second one is something that I see too often: there is no access to or descriptions of data analysis details at all. For example, the github portal, or similar, would be a good way to show the code and how analyses were conducted. We usually do not accept laboratory reports without sufficient descriptions of methods (not always very detailed, agreed!), in the same way reports heavily based on data analysis should show how the data analyses were done.

Authors’ reply: The authors thank and agree with the comment. We have uploaded all the codes involved in this study to a github folder https://github.com/RuiCao34/PRS_Kidney_AR. Unfortunately, due to a consent limitation, the phenotype and genotype data in the DeKAF Genomics cohort are not publicly available. Data from the GEN-03 cohort are available through dbGaP. We also annotated the code to ensure its reproducibility.

---

## [Decision Letter · Decision Letter 1]

25 Apr 2024

Polygenic Risk Score for Acute Rejection Based on Donor-Recipient non-HLA Genotype Mismatch

PONE-D-23-29389R1

Dear Dr. Guan,

We’re pleased to inform you that your manuscript has been judged scientifically suitable for publication and will be formally accepted for publication once it meets all outstanding technical requirements.

Kind regards,

Prof. Pierre Bobé

Academic Editor

PLOS ONE

Reviewers' comments:

Reviewer's Responses to Questions

**Comments to the Author**

1. If the authors have adequately addressed your comments raised in a previous round of review and you feel that this manuscript is now acceptable for publication, you may indicate that here to bypass the “Comments to the Author” section, enter your conflict of interest statement in the “Confidential to Editor” section, and submit your "Accept" recommendation.

Reviewer #2: All comments have been addressed

2. Is the manuscript technically sound, and do the data support the conclusions?

Reviewer #2: Yes

3. Has the statistical analysis been performed appropriately and rigorously? 

Reviewer #2: Yes

4. Have the authors made all data underlying the findings in their manuscript fully available?

Reviewer #2: Yes

5. Is the manuscript presented in an intelligible fashion and written in standard English?

Reviewer #2: Yes

6. Review Comments to the Author

Reviewer #2: Thank you for the amendments you have done to the revised version of the manuscript, the manuscript now is fine to me.

7. PLOS authors have the option to publish the peer review history of their article (what does this mean?). If published, this will include your full peer review and any attached files.

Reviewer #2: **Yes: **Partanen, Jukka

---

## [Editor Report · Acceptance letter]

20 May 2024

PONE-D-23-29389R1 

PLOS ONE

Dear Dr. Guan, 

I'm pleased to inform you that your manuscript has been deemed suitable for publication in PLOS ONE. Congratulations! Your manuscript is now being handed over to our production team.

Kind regards, 

on behalf of

Prof Pierre Bobé 

Academic Editor

PLOS ONE